# Low-Slow-Small (LSS) Target Detection Based on Micro Doppler Analysis in Forward Scattering Radar Geometry

**DOI:** 10.3390/s19153332

**Published:** 2019-07-29

**Authors:** Surajo Alhaji Musa, Raja Syamsul Azmir Raja Abdullah, Aduwati Sali, Alyani Ismail, Nur Emileen Abdul Rashid

**Affiliations:** 1Wireless and Photonics Networks (WIPNET), Department of Computer and Communication System Engineering University Putra Malaysia (UPM), Serdang 43400, Selangor Darul Ehsan, Malaysia; 2Computer Engineering Department, Institute of Information Technology Kazaure, Kazaure 5002, Jigawa, Nigeria; 3Faculty of Electrical Engineering, University Teknologi Mara, Shah Alam 40450, Selangor, Malaysia

**Keywords:** micro Doppler, forward scatter radar (FSR), Low-Slow-Small (LSS) target detection

## Abstract

The increase in drone misuse by civilian apart from military applications is alarming and need to be addressed. This drone is characterized as a low altitude, slow speed, and small radar cross-section (RCS) (LSS) target and is considered difficult to be detected and classified among other biological targets, such as insects and birds existing in the same surveillance volume. Although several attempts reported the successful drone detection on radio frequency-based (RF), thermal, acoustic, video imaging, and other non-technical methods, however, there are also many limitations. Thus, this paper investigated a micro-Doppler analysis from drone rotating blades for detection in a special Forward Scattering Radar (FSR) geometry. The paper leveraged the identified benefits of FSR mode over conventional radars, such as improved radar cross-section (RCS) value irrespective of radar absorbing material (RAM), direct signal perturbation, and high resolutions. To prove the concept, a received signal model for micro-Doppler analysis, a simulation work, and experimental validation are elaborated and explained in the paper. Two rotating blades aspect angle scenarios were considered, which are (i) when drone makes a turn, the blade cross-sectional area faces the receiver and (ii) when drone maneuvers normally, the cross-sectional blade faces up. The FSR system successfully detected a commercial drone and extracted the micro features of a rotating blade. It further verified the feasibility of using a parabolic dish antenna as a receiver in FSR geometry; this marked an appreciable achievement towards the FSR system performance, which in future could be implemented as either active or passive FSR system.

## 1. Introduction

The low cost, off the shelf availability, and operational flexibility of a drone lead to its misuse by civilian society, apart from military applications, thereby increasing posing threats to society [1]. Potential threats, such as drugs smuggling, transportation of contraband materials like weapons to criminals in prison, unauthorized imaging and filming in restricted areas, illegal surveillance, air collision, terrorist attacks, and radio frequency (RF) jamming are among the potential threats posed by the drone misuse. This has become a matter of importance due to the alarming threat to our community and the recorded security breach [2] caused by a drone. It may have posed a threat to scenarios, such as (i) our recreational centers like a stadium, (ii) strategic and critical infrastructure or secured environment, etc., such that no sound can be noticed and no warnings may be provided.

Several attempts made to detect the drone, such as radio frequency-based (RF) [3,4], acoustic [5], video imaging [6], audiovisual [7], software-based smartphone [8], and other non-technical methods like shooting and netting [9], reported their achievements in various capacities. Considering the drone-generated audio frequencies that are usually within 40 kHz, drone detection may fail due to a higher noise ratio in urban cities [1]. The drone’s nature of following a predefined GPS route [4,10] provides no RF link to trace and, as such, cannot be detected. The constant update failure of the drone’s signatures needed by a referenced database in RF-based method [4] and the noisy nature of unlicensed WIFI RF bands [11] due to many users are other challenging issues. In the case of a camera-based method, detection may be hidden due to dynamic target background which, in turn, suffers few pixels representation; these make it difficult to differentiate a drone from flying birds [1,10]. The thermal technique is considered inefficient due to drone’s plastic frames and minimal heat exhaust [1].

In contrast, these make a radar system as an alternative as it does not suffer from environmental effects, such as dark, noisy, and blurred or misty environments [11]. The present challenge of detecting a target with low monostatic radar cross-section (RCS) like the drone by the modern radars [12] is still challenging due to the target size and non-reflective material. The drone is considered difficult to be detected among other biological targets, such as insects and birds existing in the same surveillance volume, and is usually outside the range of conventional radars. Besides, some military unmanned aerial vehicle (UAVs) are made with stealth technology of high absorption coefficient, resulting in low monostatic RCS. Recently, a step-frequency continuous wave (SFCW) radar was proposed for Doppler compensation, especially when detecting target like UAV. Despite the achievement made, there are some unresolved challenges the drone feature due to the rotating blades. This has become challenging due to the nature and size of the drone; thus, this paper employed forward scattering radar (FSR) system to detect a drone and utilized its micro-Doppler formed by the drone rotating blades for its identification.

An FSR is a special bistatic mode when a bistatic angle is near to 180°. In FSR geometry, the target crossing between the transmitter (Tx) and receiver (Rx) blocks the electromagnetic (EM) wave traveling from the Tx-to-Rx; this forms a target’s shadow (silhouette) irrespective of the target’s material. The FSR minimizes the effect of radar cross-section (RCS) of the target while detection [13,14]. The high target’s RCS irrespective of the target’s shape and radar absorbing material (RAM) makes it popular over the traditional monostatic radar [15]; instead, FSR considers the target physical dimension and the wavelength during detection. The FSR system has an improved feature of being counter to stealth technology, low RCS targets detection, high power yield [16], among others. These identified benefits may fit its usage in detecting a target characterized as a low altitude, slow speed, and small RCS (LSS) target [17] like a quadcopter commercial drone.

This paper aimed to conduct a micro-Doppler feature extraction and analysis of a drone rotating blades in addition to the main Doppler by using FSR geometry. We have presented a theoretical signal model, simulation, and experimental validation of the rotating blade signature. The actual drone signature in FSR geometry was presented with an extract of the LSS micro-Doppler feature, relative to the micro motions due to the blade rotation, as this can be a basis for drone classification from other similar targets. The paper is organized as follows: Section 2 describes a theoretical signal model of Doppler of the main body and the micro-Doppler supported by simulation results. Section 3 explains the experimental set-up and the experimental campaign. Section 4 illustrates the experimental results and findings, and the paper is concluded by summarizing the outcomes of the entire work, suggesting some potential future research area.

## 2. Received Signal Model in FSR

A micro-Doppler signature from a drone rotating blade was a required parameter for detection in this paper. The rotating parts of the drone may have micro-motions referenced to the main silhouette-generating center of the target’s body; this resulted in an additional frequency modulations of the reflected wave and, hence, micro-Doppler [18]. The signature is now determined by the additional motion of the target that is different from the main body’s motion [17]. The micro-motion may be caused by rotation, vibrations, or both [19,20], resulted from multiple combinations of point scatterers [21].

The FSR geometry as presented in Figure 1, had the target moving toward the baseline. The scattered signal caused an additional modulation due to the rotating blades, thus, forwarded to the receiver as received signal.

From Figure 1, a combination of both the scattered and the direct signal forms the received signal S_rx_ = S_dir_ + S_sc_. Assuming the direct signal is given by S_dir_ = A_dir_Cos(ω_c_)t, and the scattered signal be S_sc_ = A_sc_Sin(ω_c_ + ω_sc_)t, the received signal, as in [22], is given by
S_rx_ = A_dir_Cos(ω_c_)t + A_sc_Sin(ω_c_ + ω_sc_)t(1)
where ω_c_ = 2πF_c_ and ω_sc_ = 2πF_d_. Equation (1) is the representation of the entire signal at the receiver. Within the scattered signal, a Doppler component is present. The received signal passes through a non-linear device and yields a signal with a square law on the output. A filtering process is then applied using a low pass filter (LPF) to filter out the high frequency component from the diode output, leaving only the two spectral components, i.e., the direct leakage signal and the signal containing the Doppler component scattered by the target and a conversion factor due to the type of diode used. The direct signal (leakage) component can be filtered out by using a high pass filter (HPF), leaving the signal with only a Doppler component, thus, ω_sc_ is the angular frequency combining the Doppler component *f_d_* due to linear motion and the micro-Doppler *f_md_* due to the blade rotation; thus, ω_sc_ = 2π*f_d_*t + 2π*f_md_*t, as described in [22], and the received signal equation is now:S_rx_(t) = A_dir_ ∗ A_sc_ ∗ Sin (2πf_d_ + 2πf_md_)t(2)
From [22], the general Doppler equation for the bistatic arrangement is given by
(3)$$f_{d}=\frac{2V}{\lambda }\cos\left(\frac{\beta }{2}\right)\cos(\delta )$$
where V is the radial velocity, β is the Tx-target-Rx bistatic angle, [$\cos\left(\frac{\beta }{2}\right)\cos(\delta )$] are the coefficient factors due to bistatic geometry, and λ is the wavelength. A detailed derivation of the Doppler equation is also in [22].

In this paper, an actual quadcopter drone geometry with 4-rotors of 2-rotating blades, maintaining similar dimensions was represented by a crossbar in FSR geometry, as illustrated in Figure 2. In the geometry, the drone was placed equidistance between the transmitter and the receiver. Two significant scenarios, such as horizontally (Facing-Up) and vertically (Facing-Rx) aligned, 1 m away from the baseline, were modeled. If the T_x_ is located at the origin of fixed coordinates (X_0_, Y_0_, Z_0_ = 0) and the R_x_ placed on the same plane of equal height separated by a baseline distance B_L_, the corresponding R_x_ coordinate is given by (B_L_, 0, 0). The rotors positions were assumed to be at the tip ends of the crossbar, and the rotor vectors signifying their positions on the (xyz) plane was defined. In the ‘Facing-Rx’ scenario, the blades of all rotors were made to rotate about the *Z*-axis, and the *X*-axis for ‘Facing-Up’ scenarios, as illustrated in Figure 2a,b. The corresponding position vectors of all the rotors for each case are described in Equations (4) and (5).

If the rotor vectors are represented by R_na_ and R_nb_, (for n = 1… 4), where n corresponds to the number of rotors, ‘a’ and ‘b’ are the corresponding scenarios for ‘Facing-Up’ and ‘Facing-Rx’, respectively, D is the distance travel, and W_d_ is the width of the drone, the corresponding rotor vectors are represented by:(4)$$\begin{array}{c} \vec{R1a}=\left[\left(\frac{B_{L}}{2}\right),\;D,\;0\right]\\ \vec{R2a}=\left[\left(\frac{B_{L}}{2}+\frac{W_{d}}{2}\right),\left(D-\frac{W_{d}}{2}\right),\;0\right]\\ \vec{R3a}=\left[\left(\frac{B_{L}}{2}\right),\;\left(D-W_{d}\right),\;0\right]\\ \vec{R4a}=\left[\left(\frac{B_{L}}{2}-\frac{W_{d}}{2}\right),\left(D-\frac{W_{d}}{2}\right),\;0\right] \end{array}$$
(5)$$\begin{array}{c} \vec{R1b}=\left[\left(\frac{B_{L}}{2}\right),\;D,\;0\right]\\ \vec{R2b}=\left[\left(\frac{B_{L}}{2}\right),\left(D-\frac{W_{d}}{2}\right),\;\left(-\frac{W_{d}}{2}\right)\right]\\ \vec{R3b}=\left[\left(\frac{B_{L}}{2}\right),\;\left(D-W_{d}\right),\;0\right]\\ \vec{R4b}=\left[\left(\frac{B_{L}}{2}\right),\left(D-\frac{W_{d}}{2}\right),\;\left(+\frac{W_{d}}{2}\right)\right] \end{array}$$

From Equations (4) and (5), R_1a_ represents rotor-1 of Figure 2a and R_1b_ refers to rotor-1 of Figure 2b. Each of this rotor serves as the center of 2-blades upon which the blades rotate. Based on the rotor positions of Figure 2, a receiver-range geometry of two rotating blades and their corresponding range profiles are presented in Figure 3. The ranges between the receiver-to-blade-tip are given by R_rx__tip_nm_, and the transmitter-blade-tip by R_tx__tip_nm_. These make a bistatic angle ‘β_tip_’ for each rotor ‘n’ blade ‘m’. Similarly, R_tx__Fixed_n_ and R_rx__Fixed_n_ are the corresponding ranges for transmitter-to-rotor fixed position and rotor-to-receiver, with bistatic angle ‘β’. The scattered signal can, therefore, be calculated based on the location and orientation of the blades.

Figure 3 is a zoomed view of only one rotor described in Figure 2. Each rotor has 2-blades rotating about a fixed rotor point that forms the bistatic angle between the receiver and the transmitter. It describes the Doppler due to the blade-tip and the blade’s range profile referenced to both transmitter and the receiver. The ‘β_tip_’ is the bistatic angle formed by the blade-tip while rotating through 0–2π, about a fixed rotor position. The variation of the blade yaw angle is determined by the rotation rate, the blade length L_b_, and the initial angle given by ‘θ’. If the blade rotates at a constant rotation rate (Ω) over time (t), a yaw angle is formed (Ωt + θ_o_), and, thus, the velocity is caused by the blade tip. The velocity of the blade-tip (V_tip_), which is a function of the Doppler generated, is indicated in (6).
V_tip_ = 2πL_b_Cos(Ωt + θ_o_)(6)
where L_b_ is the blade length, 2π is the conversion rate of radial velocity. The received range reference to the blade tips can be obtained by using (7)
R_rx__tip_nm_ = [R_rx__Fixed_n_^2^ + L_b_^2^ − (2 ∗ R_rx__Fixed_n_ ∗ L_b_) ∗ Cos(θ + θ_n_)]^1/2^(7)
where R_tx__Fixed_n_ and R_rx__Fixed_n_ are the corresponding transmitter and receiver ranges for the rotor ‘n’. These ranges are used to obtain the bistatic angle ‘β’ for the target’s radial motion. Similarly, R_rx__tip_nm_ and R_tx__tip_nm_ are the corresponding ranges of receiver-to-blade-tip and transmitter-to-blade-tip, for rotor ‘n’, blade ‘m’. These are also used to calculate the bistatic angle ‘β_tip_’ made by the blade-tip at each point while rotating. From (2), the total received signal for rotor ‘n’ with ‘N’ maximum number of blades, the sum of the received signal is, therefore, represented as:(8)$$S_{\mathrm{rx}}(t)=\sum_{k=0}^{N-1}\mathrm{Arx}\left(n\right)\mathrm{Sin}\left[2\pi \mathrm{fdt}+\Phi_{\mathrm{md}}n\left(t\right)\right]$$
Φ_md_(t) is the phase modulation generated as a result of a circumferential micromotion of each blade and can be represented as
(9)$$\Phi_{\mathrm{md}}\left(t\right)=-\frac{2\pi }{\lambda }\frac{L_{b}}{2}\cos\left(\frac{\beta }{2}\right)\cos\left(\delta \right)\cos\left(\Omega t+\theta o\right)$$

L_b_ is the blade length, Ωt and θ_o_ are the rotation rates of the rotating blade in the rs−1 and initial angle of the blade, respectively. If a rotor has multiple numbers of blades, the rotating scatterers of each blade with different initial angles [23] can be generated by using θ_n_ = θ_o_ + n2π/N, for n = 0, 1, …, N − 1; in our case, the rotor had only 2-blades, therefore, n = 0, 1.

As shown in Figure 3, the angle ‘θ_o_’ was assumed to lie at the origin, and this made the initial angle ‘θ_o_’ to be zero [18], leaving only the blade rotation rate Ωt. We used the Bessel function of the first kind to obtain the micro-Doppler frequency of each blade. This was achieved by differentiating Equation (9) for time ‘t’ [24,25].
(10)$$f_{md}=\frac{1}{2\pi }\frac{d}{\mathrm{dt}}\delta \mathrm{md}\left(t\right)$$
(11)$$=\frac{1}{2\pi }\frac{d}{\mathrm{dt}}\left(-\frac{2\pi }{\lambda }\frac{L_{b}}{2}\cos\left(\frac{\beta }{2}\right)\cos\left(\delta \right)\cos\left(\Omega t+\theta n\right)\right)$$
(12)$$f_{md}(\max)=\frac{\mathrm{Vtip}}{\lambda }\cos\left(\frac{\beta }{2}\right)\cos\left(\delta \right)$$
where V_tip_ is the velocity of the blade tip. If 2π is the conversion rate of the angular speed Ω_t_, L_b_ is the blade’s length, then V_btip_ = 2π ∗ L_b_ ∗ Ω_t_. The maximum micro-Doppler is now represented as
(13)$$f_{md}(\max)=\frac{2\pi L_{b}\;x\;\Omega }{\lambda }\cos\left(\frac{\beta }{2}\right)\cos\left(\delta \right)$$

The total signal received for ‘N’ number of blades can, therefore, be represented as
S_rx_(tot) = A_dir_ ∗ A_sc_(n) ∗ Sin(2πf_d_ + Փ_md_n)t(14)
If the received amplitude A_rx(n)_ is represented as A_r_ ∗ A_e_(n), then the total received signal is
(15)$$S_{\mathrm{rx}}(\mathrm{tot})\sum_{k=0}^{N-1}A_{\mathrm{rx}}(n)\mathrm{Sin}(2\pi f_{d}+\Phi_{\mathrm{md}}n)t$$

In summary, the received signal of each of the four rotors can be treated independently based on their geometrical positions.

Based on the described geometry of Figure 2 and Figure 3, Equation (12) was used to generate the Doppler due to blade rotation with ‘V_tip_’ as velocity due to blade rotation. The bistatic angle made by the blade-tip ‘β_tip_’ was used in the Doppler equation; hence, the Doppler equation was $f_{md}=\frac{V_{\mathrm{tip}}}{\lambda }\cos\left(\frac{\beta \mathrm{tip}}{2}\right)\cos\left(\delta \right)$. A rotating speed of 36 r/s, three (3) continuous wave (CW) frequencies, i.e., 3 GHz S-band (for air traffic control), 5 GHz C-band (for weather radars), in addition to 7 GHz, were used to demonstrate the simulated micro-Doppler of the rotating blades. Each frequency used satisfied the equation 2πƖ/λ >10 in a way that the target dimension was electrically large when compared with the wavelength [26], where ‘Ɩ’ is the target radius. These frequencies were also used during the experimental micro-Doppler analysis. The corresponding short-time Fourier transform (STFT) signature representation is illustrated in Figure 4. Figure 4a–c resulted from Facing-Rx geometry of Figure 2b, while Figure 4c,d is the signature due to Facing-Up scenario of Figure 2a.

A micro-Doppler due to the rotation of each blade-tip, while the rotor was in ‘On’ position, for an observation time of 1 s, is presented in Figure 4, with both blades preceding one another. The signal response of each blade was generated independently, based on the geometrical position to their corresponding rotor position. The signature of each blade was represented as R_11_, R_12_, R_31_, R_32_ with only rotor R_2_ and R_4_ mapped into one another. It was observed that the Facing-Up scenario yielded better resolution by separating the two mapped rotors R_2_ and R_4_, as can be seen in Figure 4d–e. Result in Figure 4 shows the resolution improved by increasing the operating frequency. This is because micro-Doppler is proportional to the operating frequency. This revealed how smaller wavelength is expected to yield a better Doppler resolution during the experimental work. Based on the simulated results, the signatures due to rotating blades corresponds to the findings in [17,19]. The observation angle played a vital role in the Doppler and range resolutions, as can be seen between ‘Facing-Up’ and ‘Facing-Rx’ blade orientation. In the ‘Facing-Up’ scenario of Figure 4d–f, the overlapping rotors R_2_ and R_4_ can be seen, unlike the ‘Facing-Rx’ case.

## 3. Experimental Setup

The experiments conducted in this paper consisted of two parts, as described in Figure 5: The first part aimed to validate the micro-Doppler theoretical model, as described in Section 2. The first part comprised of an experiment in the anechoic chamber and outdoor experiment with the low elevation of the transmitter and FSR receiver.The second part aimed to study the feasibility of the proposed FSR system to detect the drone’s linear motion and to extract the micro-Doppler of the rotating blade.

Figure 5 describes the general composition of the experiments conducted in this study.

For the validation, the experiment was conducted with fixed rotating blades for outdoors and in the anechoic chamber.

### 3.1. Signal Processing and Receiver Hardware

The FSR has simple receiver circuits that do not require any synchronization channel and, thus, contains only one receiving channel with a self-mixing technique using a non-linear amplitude detector (Diode). The signal arriving the receiving antenna is composed of a superposition of two signals, the direct path signal from the transmitter and the echoed signal by the target. The signal composition is directly fed into the low noise amplifier (LNA). The amplitude detector performs a demodulation process on the amplified signal from the LNA output and then extracts a very low-frequency band from the phase-amplitude modulated signal. The low pass filter (LPF) outputs the very low Doppler signal content and discards the high-frequency component. This signal was saved on the PC for further processing. Figure 6 describes the hardware composition of the FSR receiver circuit. During the experiment, the signal sampling frequency of *f_s_* = 50 kHz knows the expected Doppler due to the rotating blades. This was further down-scaled by 10 to *f_s_* = 5 kHz for unambiguous generated Doppler during the offline Matlab signal processing. For the rotating blade signature, a clear description between the ‘On’ position of the rotating blade, the ‘Off’ state, and their corresponding transition states is presented in all cases. For drone detection, the same signal processing was maintained after the signal was received.

## 4. Experimental Results

In this study, Section 4.1 and Section 4.2 focus on the “micro-Doppler” components due to the rotating blades and are compared with the simulation. The experiment in Section 4.3 considered both the main Doppler due to drone linear motion and the micro-Doppler due to the blade micro motions. The experiment is more practical due to the baseline range, location of the transmitter, and antenna specifications. For Section 4.1 and Section 4.2, the experiment used “horn antenna” at both transmitting and receiving end, while in Section 4.3, the 11 GHz experimental setup involved a horn antenna transmitter and a parabolic dish receiver. In Section 4.3, we considered using 11 GHz due to the following reasons:(a)To have a more meaningful and practical FSR system, future work will involve passive system by using illuminator of opportunity. For drone detection using FSR geometry, the best illuminator is using a satellite signal; thus, the proposed 11 GHz in our study could resemble satellite signal working at K/Ka/Ku band.(b)Based on the satellite signal as mentioned in (a), the system needs to use a dish antenna for acceptable Signal to Noise (SNR). Thus, practically using lower frequency will make the dish antenna bigger, so at 11 GHz, the antenna size is reasonable.(c)To verify the performance of our developed receiving system that only operates within Ku-band frequencies, in the detection of a real commercial drone.

The experiment aimed at detecting and extracting the micro-Doppler of rotating blades. The blade is 0.12 m length, which made to rotate at an approximate speed of 36 r/s. Firstly, the blades are wrapped with a conducting material (usually aluminum foil) to improve its reflective property, because most drones are made of either plastic material or carbon fiber [27]. Subsequently, the default blades are used without been wrapped to determine the degree of their detectability as compared to the first case. During the experiment, scenarios involved in both cases include blades Facing-Receiver, Facing-Up, and maneuvered motion. The effect of look angles of the drone blade on the rotating blade while detection in addition to wavelength selection was also considered. This paper compared the simulated and the experimental signature for 3 GHz operating frequency, as illustrated in Figure 7.

Figure 7 is the STFT representation of the experimental micro-Doppler in Figure 7a and the simulated in Figure 7b. The experimental result comprised of the three-state transition of the blade, i.e., ‘Start’, ‘On’, and ‘Stop’. A three-bar signature represented the ‘On’ state of the rotating blade. The three separated bars represented the signatures of R_1_, R_2_–R_4_ (R_2_ and R_4_ mapped on one another), and R_3_, respectively. The ‘start’ and ‘stop’ states represented the blade’s transition when switched ‘On’ and ‘Off’, respectively. In both cases, blade-1 and blade-2 mapped together. Similarly, the simulation represented only the ‘On’ position of the rotating blade and had R_11_, R_12_, R_31_, R_32_, represented with R_2_ and R_4_ mapped together, as described earlier. Besides, unlike the experimental signature that represented the signature of the entire blade length, the simulated signature only represented the blade-tips. The variation in the Doppler value resulted due to multiple factors, such as the difference in obtaining the exact value of the theoretical parameters like the bistatic angle, aspect angle, blade rotation rate, etc., and the signal degradation due to other external factors.

### 4.1. Rotating Blades Outdoor

An outdoor experiment was conducted to ascertain the detectability of the drone rotating blades. It further verified the look angle effects and the effects due to the maneuvered motion of the drone. To achieve these, we used 3 GHz frequency for both Facing-Receiver and Facing-Up scenarios for illustration purposes. This was based on the observed similarity in the signature for all the three operating frequencies earlier used. Subsequently, the experimental setup, shown in Figure 8, was not to scale but rather for illustration purposes to provide insightful information on how the experiment was conducted.

#### 4.1.1. Blade Detection

Total observation time of 15 s was used to verify the detectability of the rotating blade, which was considered enough to record the three-states transition of the blades, i.e., Off-On-Off. The blade was initially in ‘Off’ position, then switched ‘On’ and then ‘Off’ again for 5 s each. Between each ‘On’ and ‘Off’ positions, a transition time was observed with a projected rise due to starting torque from ‘steady’ state to ‘rotation’ state, as seen in Figure 9. A similar occurrence happened when the rotors slowed down to steady state after it was turned ‘Off’. Both time domain and the STFT represented only 9 s because the transition time of “2 s” out of “5 s” for both start and stop were captured, while the remaining “5 s” was for “On” state. Despite the weather and other effects, such as ground reflection, clutter, and volume scattering, the detected signature was considered a success; hence, ascertained our prior claims and assumption in the designed theoretical model.

#### 4.1.2. Aspects of Angle Effect

The effect of aspect angle due to blades orientation was verified by using the two scenarios, i.e., the blades ‘Facing-Rx’ and the blades ‘Facing-Up’ orientations, as described in Figure 5. This was described by using 1-rotor, 2-blades and 2-rotors, 2-blades to display Doppler resolutions. Maintaining similar operating frequency of 3 GHz, the detected signatures are illustrated in Figure 10.

A short-time Fourier transform (STFT) signature representation of 1-rotor, 2-blades and 2-rotors, 2-blades, with the rotating blades facing the receiver is shown in Figure 10a,b. Similarly, with the rotating blades Facing-Up, their signatures can also be seen in Figure 10c,d. Two bars represented the 2-rotors signature (R_1_ and R_3_) during ‘On’ position, with blade 1 and 2 mapped on one another. It was observed that the ‘Facing-Rx’ orientation had higher power intensity as compared to the ‘Facing-Up’ scenario. In both cases, good Doppler resolutions were observed as both rotors signature were seen.

#### 4.1.3. Maneuvering Effects

Based on the drone operational behavior, we considered the blade signature due to maneuvering motion. At first, the blade was made to be in ‘On’ state facing the receiver with blades rotating at an approximate speed of 36 r/s with 3 GHz operating frequency. The entire 4-rotors would then be turned slowly about a fixed position in a clockwise direction over a complete circle 0–2π. The observed change in a signature over 360° was observed in both time domain and spectrogram representation, as shown in Figure 11. The first part of the time domain was when the blade was facing in the direction of the receiver, i.e., perpendicular to the baseline. The blades turned by 90° to face down, then faced the transmitter, and finally faced up. The signal power seemed to always be of higher intensity when facing the receiver or transmitter but degraded when facing up or down. This signature might be of help in recognizing the drone maneuvering motion in the course of actual drone flight operation.

### 4.2. Anechoic Chamber

In an attempt to minimize other external factors, a controlled environment was used to further confirm the system capability. This experiment was conducted in an anechoic chamber and focused on the feasibility of detecting a default, unwrapped plastic material blade, maintaining the same geometry. Two frequencies (5 and 7 GHz) were used in the micro-Doppler analysis for the experiment in the anechoic chamber. Based on the simulation result, only two higher frequencies were selected because this experiment aimed to validate the theoretical model.

Figure 12 illustrates the experimental setup in an anechoic chamber, showing the Tx-to-Target and Target-to-Rx geometry. Following similar scenarios, the blades were placed at the center of the baseline, with ‘Facing-Up’ and ‘Facing-Rx’ scenarios, respectively. At first, the detection was conducted with blade wrapped with aluminum foil and later repeated with the default blade without foil.

#### 4.2.1. Doppler Separation between Rotors

An essential component of this work is the detection of a drone through its blade generated micro-Doppler. It is, therefore, of great importance to have an outstanding Doppler resolution that can separate signatures based on the number of rotors used. To achieve this, we considered using 1-rotor, 2-rotors, and 4-rotors, respectively, with 5 GHz operating frequency. This may indeed help in identifying a drone from a flapping bird’s wing during classification. Similarly, a 15 s integration time with three-states Off-On-Off transitions was observed. The recording started while rotors were in ‘Off’ position for 5 s, then switched ‘On’ for another 5 s. The ‘Transition’ time between ‘Off’ and ‘On’ was represented by the projected rise in the signature due to the starting torque from ‘Off’ to ‘On’ position. Based on the number of the rotors used during ‘On’ state, Figure 13 shows a 1-bar for 1-rotor, 2-bars for 2-rotors, and 3-bars for 4-rotors, respectively, manifesting that the Doppler resolution over a reasonable distance was achieved. As always the case, the middle bar of Figure 13c represents rotor R_2_ and R_4_ as their symmetrical positions make them mapped onto one another.

#### 4.2.2. Plastic Blade Detection

Based on the facts that the default drone blades are either a plastic (usually transparent) or carbon composite material and considering their poor reflective characteristics with conventional radars, we aimed at implementing the FSR geometry coupled with performance enhancement of the chamber, to verify the detection possibility of the default plastic blade by using 5 and 7 GHz frequencies for both Facing-Rx and Facing-Up scenario. By using similar operating frequency, 1-rotor, 2-rotors, and 4-rotors were detected, as shown in Figure 14.

Despite the inverse relation of signal amplitude to the transmitting frequency, the Doppler value improved in the ratio of 10 Hz, as seen in Figure 13. The detection of default plastic blade could be considered successful, despite the loss in resolution and power intensity degradation when compared to blade wrapped with foil.

### 4.3. Actual Drone Detection

An off the shelf commercial drone was used to ascertain its detectability using the FSR geometry. This setup used a parabolic dish antenna as a receiver to increase gain. For a smooth flight condition of the drone, the transmitter was placed on the 11th floor of a tower at engineering faculty Universiti Putra Malaysia (UPM), with 40 m height (H). The transmitter was made to face an open area behind the building during the conduct of the experiment. The transmitter comprised a signal generator and a dual-polarized high-frequency horn antenna with 15 dB gain. The signal generator transmitted 11 GHz frequency with a maximum 15 dBm signal power.

The receiver was set to point towards the direction of the transmitter main lobe. For maximum power received, an effort was made to maintain the line of sight with the transmitter while observing the signal strength by using a spectral analyzer. Figure 15 illustrates the experimental geometry and the aerial view of the experimental site. The receiver consisted of a 65 cm parabolic dish antenna, with a feed horn mounted to point the focal point of the dish, to receive the focused reflected energy by the dish. The receiving antenna had a minimum gain of 36 dBi, 2.6° beam width at 3 dB points. For our purpose, the low noise block (LNB) circuit inside the feed horn was developed to suit our purpose. At first, we used a high-resolution ADC device to confirm the signal reception before the detection, as shown in Figure 15b. Based on the signal power received of (−47 dBm), we required about 20 dB gain to meet the sensitivity level of the amplitude detector for the down-conversion process and the Doppler signature extractions. To achieve this, a fabricated LNA operating within Ku-band frequencies was, therefore, used during the experiment. We tried as much as possible to imitate the Satellite by using low power transmit and also the actual satellite dish receiver with a narrow beamwidth.

Figure 15 describes the experimental FSR geometry and the aerial view of the experimental site. Based on the geometry, shown in Figure 15a, the detection was conducted, as described in the subsequent sections.

#### 4.3.1. Target Detection

Before the experimental validation, the main Doppler due to the drone linear motion was simulated by using the geometry in Figure 15a. This helped in having prior knowledge of the expected signature of the target. A target linear motion was simulated with a radial velocity of 16 m/s similar to the actual drone speed. The resultant signature is depicted in Figure 16.

Figure 16 shows the simulated Doppler signal in FSR due to target linear motion based on the geometry shown in Figure 15a. Both the time domain and STFT representations depicted the phase change at the point the target approached and left the baseline; hence, an expected signature due to FSR geometry was established.

During the experimental work, we had a collection of clutter which was identified as “low clutter” environment. The clutter was further reduced by using a narrow beamwidth of the receiver antenna. Besides, the receiver was Facing-Up (to the sky) with nothing within the coverage area. Figure 15b shows the transmitter, target, and receiver positions and their formation of the FSR geometry. The drone was made to fly up and cross the baseline formed by the Tx–Rx. Immediately, the drone crossed the baseline, it blocked the signal and caused an additional modulation, which would then be propagated to the receiver. The signal, arriving the receiver, was received by the developed LNB of the receiving system. The received signal followed a conventional FSR receiver processing technique, discussed earlier in Section 3.1, and, therefore, extracted the Doppler signature, as shown in Figure 17.

Figure 17 shows how the drone flies vertically to cross the baseline with the time domain, describing the point at which the target is crossing the baseline in Figure 17b. The drone was made to hover and cross the baseline vertically with target-receiver ranges (R_rx_) for up to 40 m. The Doppler signature of the drone started as soon as the drone approached the baseline; when the bistatic angle was within the FS mode until the Doppler was approximately close to zero, while the drone was crossing the baseline. All the three-states approaching, crossing, and leaving of the drone signature can be seen in Figure 17, and the phase changes due to target crossing. The difference in the Doppler value between the simulation and the experiment might have resulted from factors, such as variation in the geometrical alignment like the aspect angle, degradation of the real signal due to mechanical noise and surrounding clutter, or affected by volume scattering and background noise. Although detection of a target crossing the baseline was established, yet, a conclusion was not made as to whether the detected target was a drone or not. To further ascertain that, an STFT was used to extract the micro-Doppler features due to blade rotation, as this could serve as an avenue to identifying the drone.

#### 4.3.2. Micro Doppler Extraction

The FSR geometry in Figure 15a was also used to simulate a micro-Doppler signature of a flying drone; this gave an insightful behavior of the combined Doppler, i.e., Doppler due to the main body and micro-Doppler due to rotating blades. As earlier discussed, the micro motions resulted in an additional modulation surrounding the main Doppler. An STFT representation of the simulated signature for one second described the expected performance of the Doppler signature when the drone crossed the baseline. This is illustrated in Figure 18.

As illustrated in Figure 18, the simulated signature provided a clear view of the Doppler due to the linear motion of the main target body of about 20 Hz and the micro-Doppler surrounding the V-shape signature spread for 1 s. Having this in mind, the experimental result can be compared to also show a combined signature of Doppler due to the drone’s body and the micro-Doppler from the rotating blades. The area covered by the signature was also within a period of 1 s as the case of the simulated signature; this is illustrated in Figure 19.

Figure 19 shows the experimental signature with micro-Doppler features in the form of spikes surrounding the main Doppler. In both the simulated and the experimental signatures, the Doppler approximately equaled to zero at the point a target crossed the baseline and tended to improve as soon as the drone moved away the baseline forming the V-shape signature of an FSR geometry, as can be seen in both cases. The simulated results complemented the experimental signature, thus confirming the detection of the commercial drone through its micro-Doppler features. Apart from target identification, the micro-Doppler feature could further be used to establish the direction of flight of the drone. In contrast, the Doppler value only reflected the speed lower than the maximum.

#### 4.3.3. Drone Direction of Flight

Here, the effect of micro-Doppler features was used for further identification of other drone movements. A drone performed different movement during flight, such as yaw, pitch, roll, maneuvering, among others, by merely altering the rotating blades mode. For instance, the drone hovered by increasing the net thrust of all the four rotors and performed the opposite when vertically descending to land. During the experiment, the observation time of 30 s was considered. The drone was made to hover after 5 s to cross the baseline height (h), stay up for a while like 20 s, and then descend to cross the baseline for the second time. The intention here was to observe the effect of both the rotating blades and the drone body, i.e., to say while hovering up; the blades were the first to approach the baseline and vice-versa for landing. Based on this scenario, a 6 s segment of the acquired signal represented a window containing the signature information, the corresponding signatures are illustrated in Figure 20.

Figure 20a shows the signature of the drone moving vertically up (hovering up) with a higher power intensity and Doppler value when the rotating blades were the first to approach the baseline, and a low Doppler value was observed as it moved away from the baseline. During the landing (Vertically down), the main drone body first approached and crossed the baseline; hence, low power intensity and Doppler value were also recorded, as shown in Figure 20b. These features could be used to identify the direction of flight, and other maneuvered motions by the drone.

## 5. Conclusions

This paper highlighted the aspiring benefits of implementing FSR geometry in detecting a target characterized as LSS target like the quadcopter drone. By using a rotating blade, the paper successfully extracted the blade’s signature for the number of rotors used, the blade orientation (aspect angle), and the drone maneuvered motion. An effort was also made for proper selection of operating wavelength to make detection easier by considering different operating frequencies. The resulting signature confirmed the assumption of a better signature with wavelength lower compared to the electrical size of the target. Before these, the paper presented a 4-rotor, 2-blade drone model in FSR geometry, which could be used for the micro-Doppler analysis of an LSS. The detection of the default plastic blade without any RCS improvement could be considered as an overwhelming achievement as compared to blades with improved RCS. The detection of an actual Phantom drone for up to an approximate practical distance of 40 m away from the receiver and the micro-Doppler feature extraction due to the blade′s micro motions might be a basis to ascertain our claim for drone detection and its micro signature analysis. Also, the benefit of detecting micro-Doppler is the capability to recognize drone from other small targets like birds, etc. Other additional effects of the micro-Doppler features identified, may form the basis of identifying other drone’s motions. This study can, therefore, be used to consider an FSR geometry a good option for LSS target detection and classification. The successful inclusion of the parabolic dish antenna as the receiver in the FSR geometry served as an avenue towards the implementation of Digital Video Broadcasting (DVB) satellite-based passive FSR system for drone detection; this has marked our current ongoing project.

## Figures and Tables

**Figure 1 sensors-19-03332-f001:**
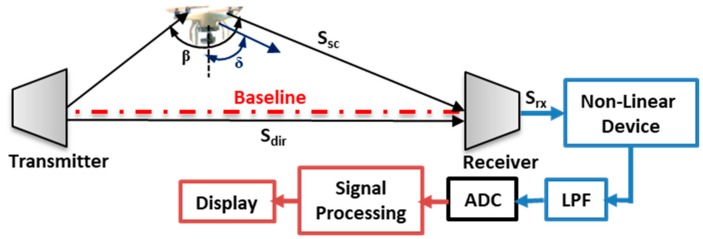
FSR (Forward Scattering Radar) geometry with receiver setup for drone detection. The receiver includes Low Pass Filter (LPF) and Analogue to Digital (ADC).

**Figure 2 sensors-19-03332-f002:**
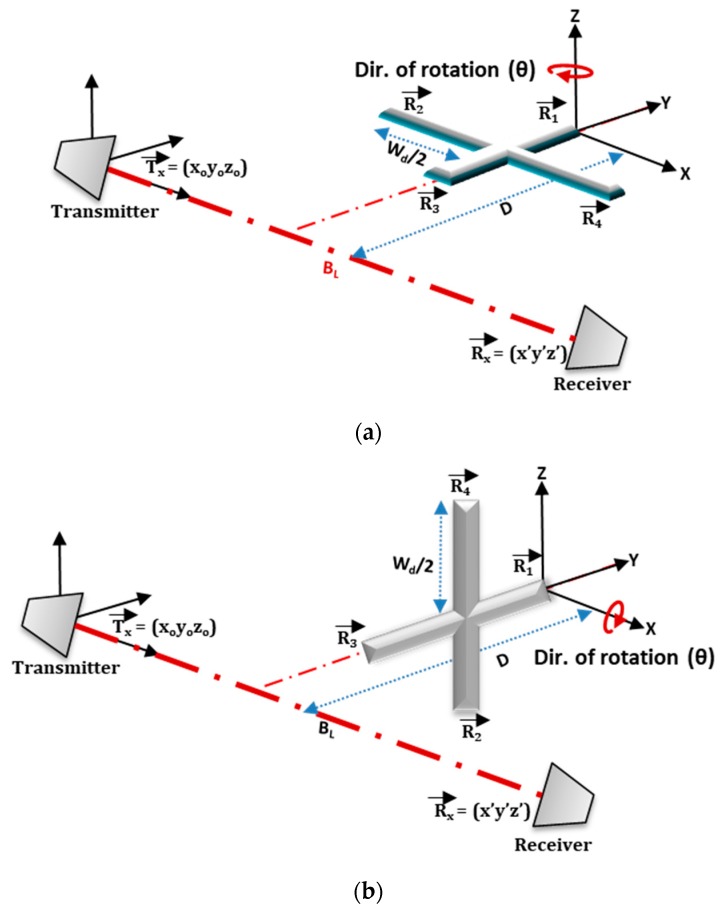
Drone blade model in FSR (Forward Scattering Radar) geometry for blade (**a**) Facing-Up and (**b**) Facing-Receiver.

**Figure 3 sensors-19-03332-f003:**
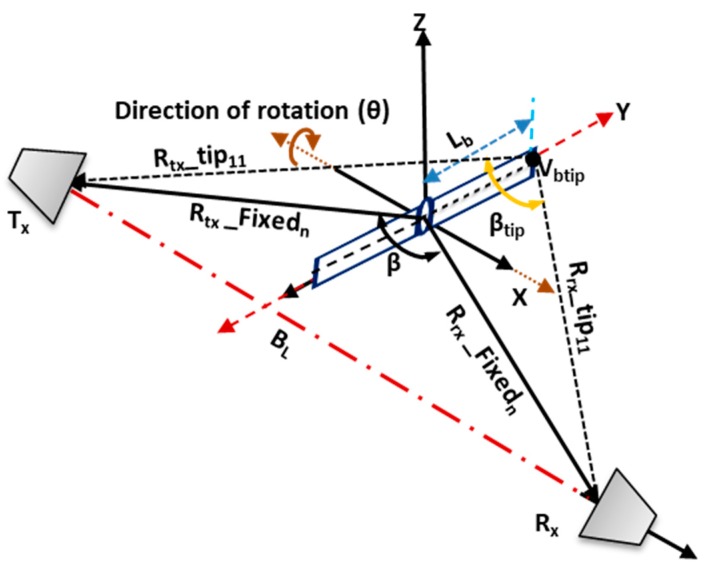
Drone rotating blades range profiles in FSR (Forward Scattering Radar) geometry.

**Figure 4 sensors-19-03332-f004:**
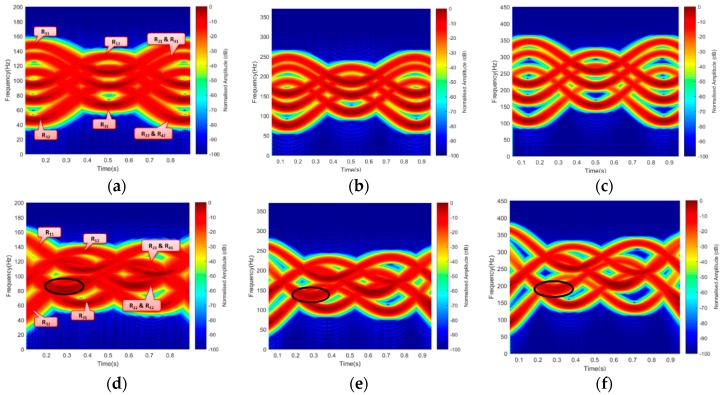
Simulated micro-Doppler from the blade for scenarios: Blade Facing-Rx at (**a**) 3 GHz, (**b**) 5 GHz, and (**c**) 7 GHz. Blade Facing-Up at (**d**) 3 GHz, (**e**) 5 GHZ, and (**f**) 7 GHz. Rx: Receiver.

**Figure 5 sensors-19-03332-f005:**
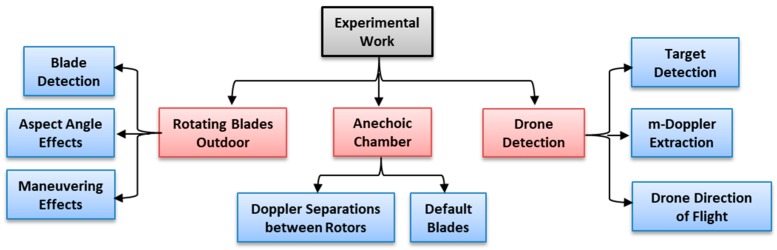
Micro-Doppler and drone detection experimental analysis and scenario set-up in FSR (Forward Scattering Radar) system.

**Figure 6 sensors-19-03332-f006:**
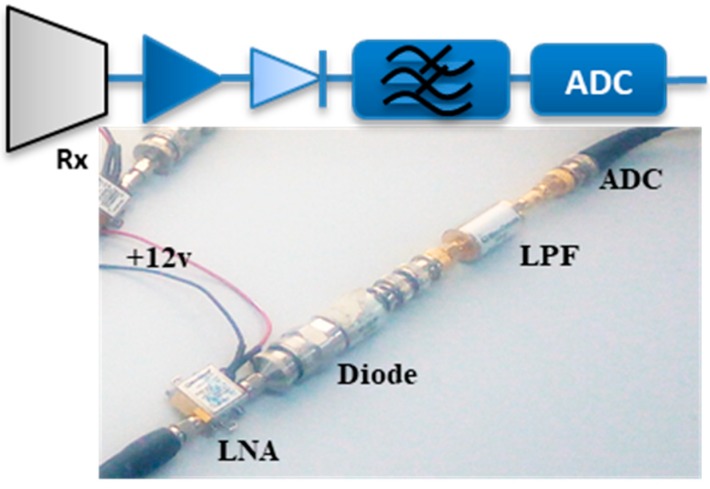
A simple FSR (Forward Scattering Radar) receiver circuit for drone detection. LNA - Low Noise Amplifier.

**Figure 7 sensors-19-03332-f007:**
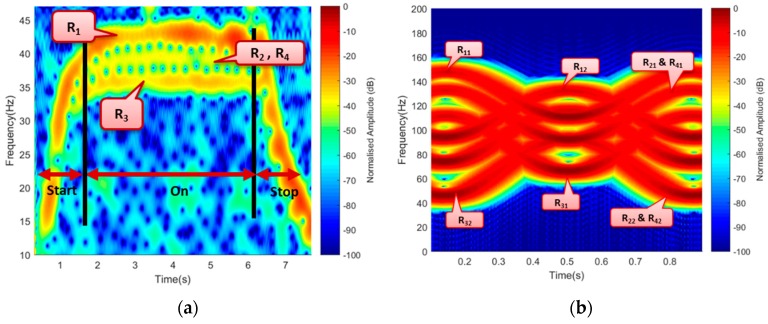
A 4-rotor, 2-blade (**a**) Experimental and (**b**) simulated Doppler.

**Figure 8 sensors-19-03332-f008:**
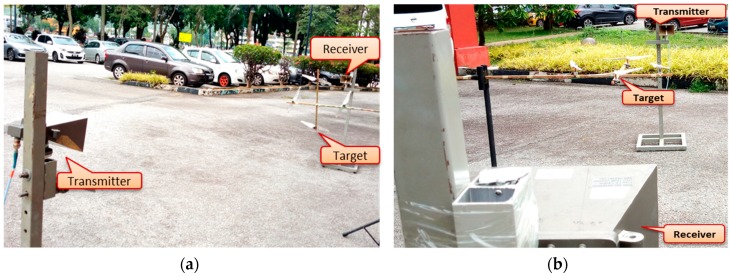
Outdoor experimental setup (**a**) Facing-Rx (**b**) Facing-Up. Rx: Receiver.

**Figure 9 sensors-19-03332-f009:**
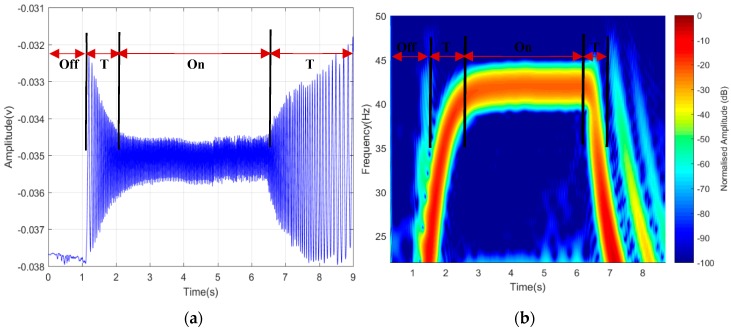
Detected signature of rotating blades in (**a**) time domain and (**b**) spectrogram.

**Figure 10 sensors-19-03332-f010:**
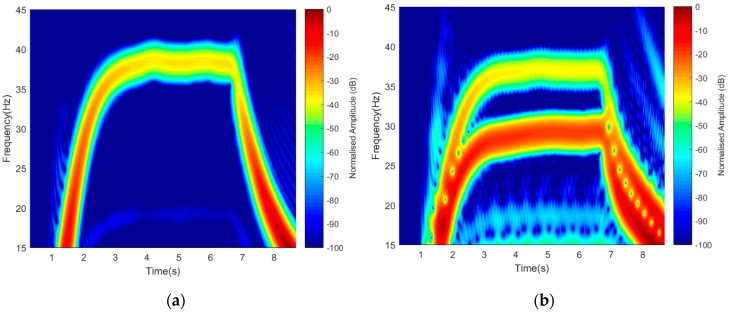
Detected signatures based on blade orientation (**a**,**b**) Facing-Rx and (**c**,**d**) Facing-Up. Rx: Receiver.

**Figure 11 sensors-19-03332-f011:**
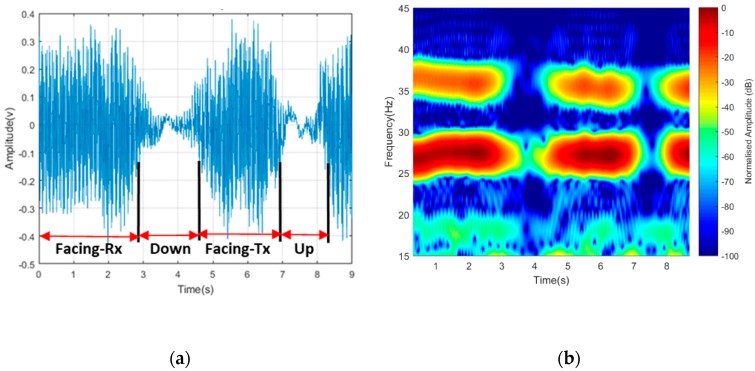
Signal from blade maneuvering in (**a**) time domain and (**b**) its spectrogram

**Figure 12 sensors-19-03332-f012:**
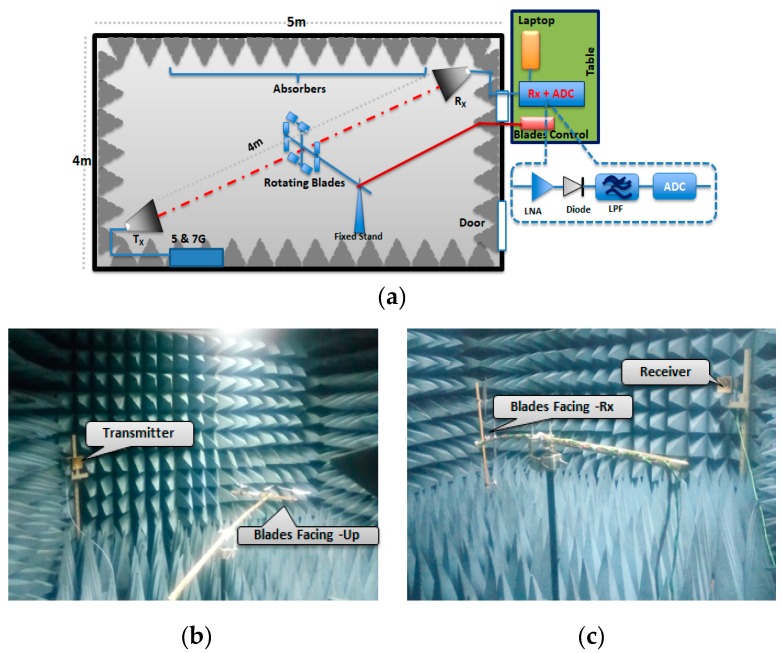
Anechoic chamber. (**a**) Experimental setup, (**b**) Blades Facing-Up, and (**c**) Blades Facing-Rx. Rx: Receiver.

**Figure 13 sensors-19-03332-f013:**
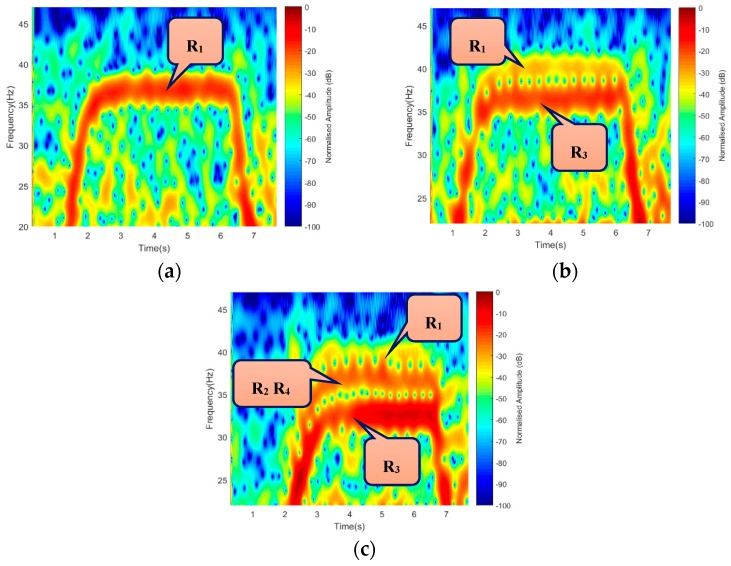
STFT (short-time Fourier transform) representation of (**a**) 1-rotor, (**b**) 2-rotors, and (**c**) 4-rotors.

**Figure 14 sensors-19-03332-f014:**
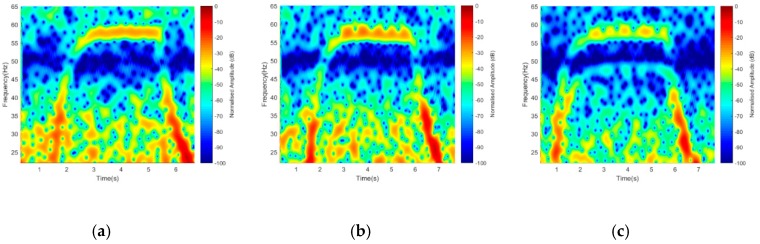
STFT (short-time Fourier transform) for default blades for 5 GHz (**a**) 1-rotor, (**b**) 2-rotors, and (**c**) 4-rotors, and 7 GHz (**d**) 1-rotor, (**e**) 2-rotors, and (**f**) 4-rotors signatures.

**Figure 15 sensors-19-03332-f015:**
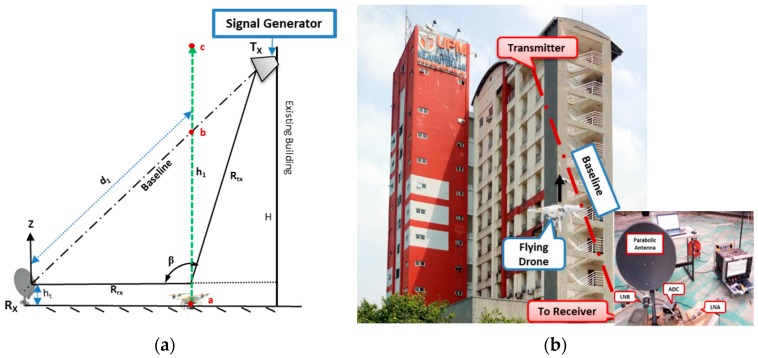
Drone detection experimental setup in (**a**) FSR (Forward Scattering Radar) geometry and (**b**) Site view.

**Figure 16 sensors-19-03332-f016:**
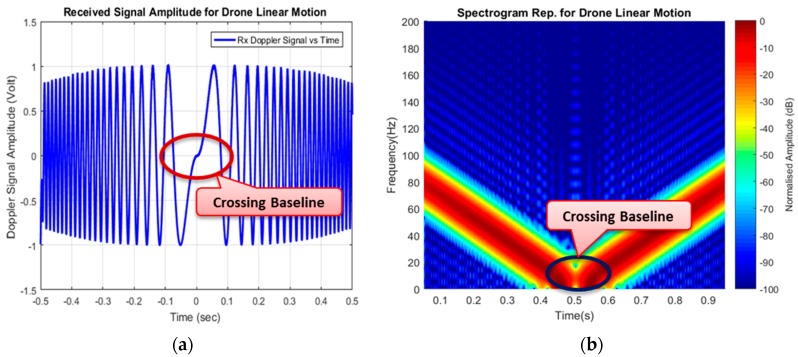
The Doppler due to target linear motion, (**a**) Time-domain and (**b**) Spectrogram.

**Figure 17 sensors-19-03332-f017:**
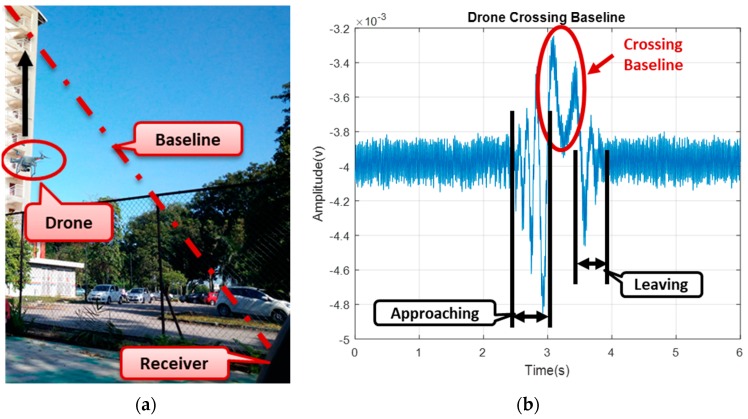
(**a**) Site view of Drone, crossing the baseline, (**b**) Raw received signal, showing the detected target’s linear motion.

**Figure 18 sensors-19-03332-f018:**
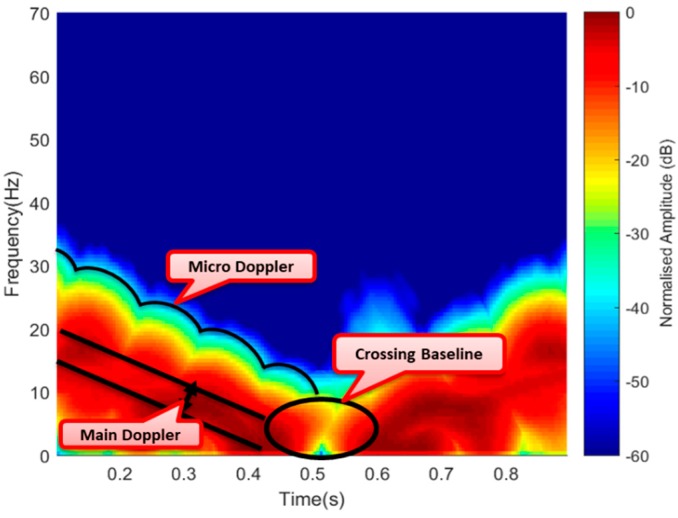
Simulated Doppler and micro-Doppler signature.

**Figure 19 sensors-19-03332-f019:**
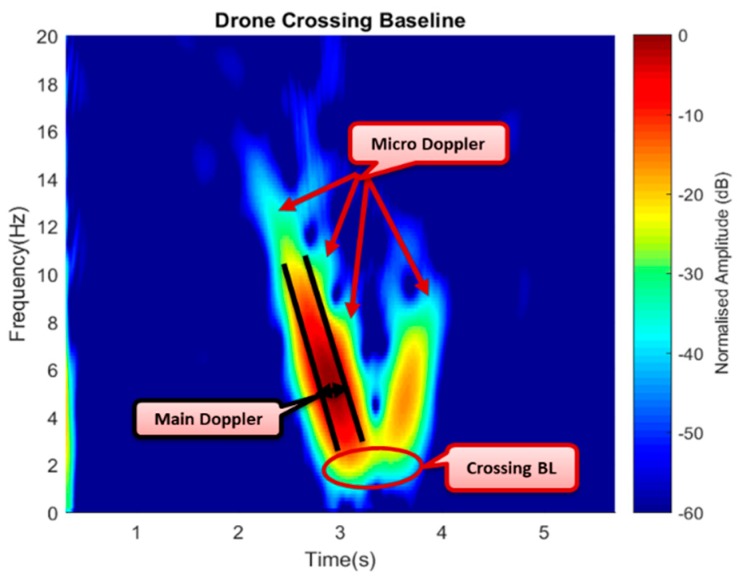
Extracted micro-Doppler from the blade and main Doppler from the body.

**Figure 20 sensors-19-03332-f020:**
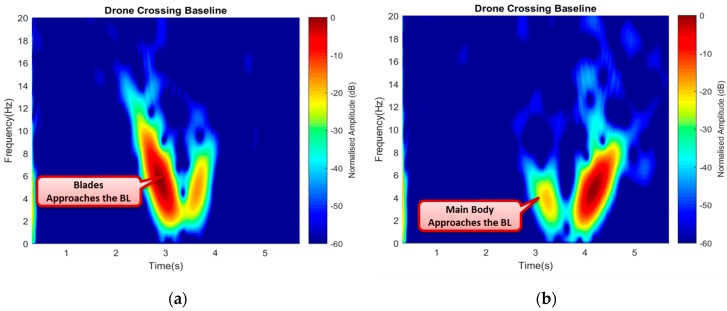
Drone crossing the baseline while moving vertically (**a**) upward and (**b**) downward.

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
