# Peer review of "Low-Slow-Small (LSS) Target Detection Based on Micro Doppler Analysis in Forward Scattering Radar Geometry"

_sensors, 2019, doi:10.3390/s19153332_

Round 1
Reviewer 1 Report
Thanks for an interesting submission. The paper provides a clear description of the FSR geometry in detecting a target characterized as LSS drone target, extracted the miro-doppler for the blade and the drone maneuvered motion. A couple of recommendations for improving the paper: a) the rotor vectors are represented in equation (3) and (4)corresponding scenarios for ‘Facing-Up’ and ‘Facing-Rx’ respectively, but it is not clear how to effect the microdoppler, although some expression in the velocity caused by the blade tip. Could it give the calculation model of the drone echo, using the numerical calculus and derivation. a) The Experimental Setup was provided in section 3, there is detail description of receiver, but not the transmitter signal. Whether is it a cos signal, OFDM or a chirp signal, and how about the bandwidth. And in section 3, it is shown a used signal generator with threes elected different frequencies, could it introduce the experimental result of microdoppler at these threes(3GHz,5GHz,7GHz), and discuss they are same or not? b) In section 4, The signal generator transmitted 11 GHz, why not give some experiment at 3GHz,5GHz,7GHz? Or say they are same. c) In section 4, the paper obtained the Blade Detection, by outdoor experiment conducted to ascertain the detectability of the drone rotating blades. And the micro-doppler feature was obvious, but how to do the cancelation for the surrounding clutter or big volume scattering, is just using the micro Doppler? Is ok under the strong clutter with the spectrum spread problem. d) In section 4, the Maneuvering effects was discussed in the bi-radar system, and the figure is good of Simulated Doppler and micro-Doppler signature, was to observe the effect of both the rotating blades and the drone body, in fact the effect of micro-Doppler features is not weak. Is there some proceeding the extract the micro-Doppler, through time frequency analysis or high-order cumulant methods e) There are some minor problems in the format that need to be corrected. The capitalization of letters is not uniform,for example ,the title of the Figure 12,(2) 2-rotors, here is (b) 2-rotors.Author Response
Response to Reviewer’s Comments
We would like to thank the editor and all the reviewers of our submitted manuscript Sensors-516766, entitled “Low-Slow-Small (LSS) Target Detection based on Micro Doppler Analysis in Forward Scattering Radar Geometry". Here we have modified the paper for resubmission according to the reviewer’s suggestions and comments.
The authors have managed to answer almost all the reviewer comments and suggestions.
REVIEWER 1
General comment
Thanks for an interesting submission. The paper provides a clear description of the FSR geometry in detecting a target characterized as LSS drone target, extracted the micro-Doppler for the blade and the drone manoeuvre motion. A couple of recommendations for improving the paper:
1) The reviewer’s comment:
“The rotor vectors are represented in equation (3) and (4) corresponding scenarios for ‘Facing-Up’ and ‘Facing-Rx’ respectively, but it is not clear how to effect the micro-Doppler, although some expression in the velocity caused by the blade tip. Could it give the calculation model of the drone echo, using the numerical calculus and derivation?”
Response:
Firstly, we would like to thank the reviewer for the comments and important suggestion, we really appreciate that. To response to this comment, we have revised the paper and the updated version of this manuscript contained a more detailed and clear explanation from line 88-313.
In the proposed FSR system, the micro-Doppler is scattered by the blade and is a function of relative velocity between the transmitter – blade – receiver. The highest rotational velocity is the blade tip and thus we included only the rotational velocity due to the blade-tip “Vtip”. For the derivation, to make it clearer, we have added new figure “Figure 1” to describe how the signal was initiated. Equation (1) Srx = Adir Cos(ωc)t + Asc Sin(ωc + ωsc)t represented the entire received signal which is a combination of direct and scattered signal. The manuscript explains further on the micro–Doppler within the received signal as describe in equation (2) – (13). This equation was used to generate a Doppler effect due to the drone radial motion; as can be seen in Fig 1 of the updated version of the manuscript.
2) The reviewer’s comment:
i. “The Experimental Setup was provided in section 3, there is detail description of receiver, but not the transmitter signal. Whether is it a cos signal, OFDM or a chirp signal, and how about the bandwidth.
Response:
Thank you for the comment. For the feasibility of the proposed FSR system to detect micro-doppler, the experiment was based on the unmodulated continuous wave signal (COS signal) as a transmitting signal for all experiments. The detail of the transmitted signal is provided in the revised manuscript.
ii. And in section 3, it is shown a used signal generator with threes elected different frequencies, could introduce the experimental result of micro-Doppler at these threes (3GHz, 5GHz, 7GHz), and discuss they are same or not?”
Response:
First of all, an explanation has been rewrite in the revised manuscript in section 3.
The three selected frequencies (3, 5 and 7 GHz) were used in the simulation analysis and the manuscript made detail comparison of the scattered micro-doppler in line 304-316.
Two frequencies (5 and 7 GHz) were used in micro-doppler analysis for experiment in the anechoic chamber. Based on the simulation result, only two higher frequencies were selected because the aim of this experiment is to validate the theoretical model.
Only 3 GHz was used in the outdoor experiment. The outdoor experimental setup requires low elevation of transmitter and FSR receiver. This condition increases the clutter and noise at higher frequencies. Thus, the low frequency at 3 GHz is sufficient for this experiment.
3) The reviewer’s comment:
“In section 4, The signal generator transmitted 11 GHz, why not give some experiment at 3GHz,5GHz, 7GHz? Or say they are same”.
Response:
The theoretical and experimental analysis which based on “3,5 and 7 GHz” is suitable and sufficient for feasibility studies. The study focuses the “micro-Doppler” components due to the rotating blades and was presented in section 4.1 and 4.2, and compared with the simulation.
The experiment in section 4.3 considered both the main Doppler due to drone linear motion and the micro-Doppler due to the blade micro motions. The experiment is more practical due to the baseline range, location of transmitter and antenna specifications. For sections 4.1 and 4.2, the experiment used “horn antenna” at both transmitting and receiving end, while in section 4.3, the 11 GHz experimental setup involved a horn antenna transmitter and a parabolic dish receiver.
Reason we considered using 11 GHz due to the following reasons
a) To be more meaningful and practical FSR system, future work will involve passive system by using illuminator of opportunity. For drone detection using FSR geometry, the best illuminator is using satellite signal, thus, the proposed 11 GHz in our study could resemble satellite signal working at K/Ka/Ku band.
b) Based on satellite signal as mentioned in (a), thus the system need to use dish antenna for acceptable SNR. Thus, practically using lower frequency will make the dish antenna bigger, so at 11 GHz, the antenna size is reasonable.
c) To verify the performance of our developed receiving system that only operates within Ku-band frequencies, in the detection of a real commercial drone.
This reason is included in the revised version line 373-389.
4) The reviewer’s comment:
“In section 4, the Manoeuvring effects was discussed in the bi-radar system, and the figure is good of Simulated Doppler and micro-Doppler signature, was to observe the effect of both the rotating blades and the drone body, in fact the effect of micro-Doppler features is not weak. Is there some proceeding the extract the micro-Doppler, through time frequency analysis or high-order cumulant methods”
Response:
The first part of section 4 only focused on the micro-Doppler of the fixed rotating blade, within the FS mode, hence, the maneuvered motion was based on the blades alone. The second part of section 4 considered both the Doppler due to the linear motion of the drone body and the micro-Doppler due to the rotating blades. For now, we only the basic procedure to extract doppler and micro-doppler frequency from the raw received signal of the drone crossing the baseline and presented it by using STFT. The raw received signal was detected by the non-linear amplitude device. The detail parameter for of STFT is included in the revised version.
5) The reviewer’s comment:
“In section 4, the paper obtained the Blade Detection, by outdoor experiment conducted to ascertain the detectability of the drone rotating blades. And the micro-Doppler feature was obvious, but how to do the cancelation for the surrounding clutter or big volume scattering, is just using the micro Doppler? Is ok under the strong clutter with the spectrum spread problem”.
Response:
We have collection of clutter which was identified as “low clutter” environment. The clutter is further reduced by using narrow beamwidth of the receiver antenna. In addition, the receiver is facing up (to the sky) where nothing within the coverage area. This explanation is included in the revised version line 718-720.
6) The reviewer’s comment:
“There are some minor problems in the format that need to be corrected. The capitalization of letters is not uniform, for example the title of the Figure 12, (2) 2-rotors, here is (b) 2-rotors”.
Response:
Firstly, we would like to thank the reviewer for this observation, we really appreciate that. The observation was noted and corrected in the updated version “now Fig 14” of this manuscript, line 646.
Regards
RSA Raja Abdullah

Reviewer 2 Report
This manuscript aims at detecting LSS target based on micro Doppler analysis in forward scattering radar geometry. This topic is very interesting, but there still have some points need to be improved.
1. In (1), Adir and Aech should be Sdir and Sech.
2. In line 186, the Ωt should be Ωt .
Author Response
Response to Reviewer’s Comments
We would like to thank the editor and all the reviewers of our submitted manuscript Sensors-516766, entitled “Low-Slow-Small (LSS) Target Detection based on Micro Doppler Analysis in Forward Scattering Radar Geometry". Here we have modified the paper for resubmission according to the reviewer’s suggestions and comments.
The authors have managed to answer almost all the reviewer comments and suggestions.
REVIEWER 2
General comment
This manuscript aimed at detecting LSS target based in micro-Doppler analysis in forward scattering radar geometry. This topic is very interesting, but there still have some points need to be improved.
1) The reviewer’s comment:
“The reviewer suggests to improve some points in the paper e.g. in (1), Adir and Aech should be Sdir and Sscat”.
Response:
Firstly, we would like to thank the reviewer for the comments and important suggestion, we really appreciate that. Yes, the reviewer is right; as there are some unclear representation of the equations as suggested by the reviewer. This was now improved in the updated version of the manuscript by introducing Fig 1 for more clear view of the FSR geometry and how the signal was initiated (see line Fig 1, line 98-110). At first Aech and Sech are now represented as the amplitude of the scattered signal ‘Asc’ and scattered signal ‘Ssc’ instead of echoed signal as earlier represented. The direct signal is given by Sdir = Adir Cos(ωc)t and the scattered signal as Ssc = Asc Sin(ωc + ωsc)t therefore Adir and Asc are the corresponding amplitudes of Sdir and Ssc. The detailed description of the equations used can be seen from line 113 to 127.
2) The reviewer’s comment:
“In line 186, the Ωt should be Ωt”
Response:
Thank you for highlighting this issue, this was noted and corrected in the updated version of the manuscript as seen in line 214.
Regards
RSA Raja Abdullah

Reviewer 3 Report
Interesting paper showing both theory and measurements.
Both the simulated and the measurement results are very interesting but they are not fully explained.
Fig 3. - The model seems to be limited to the blade tips (and it should be clearly stated). In many cases also blade flashes are visible. Electromagnetic modeling will be welcome
Fig 7. Do we observe Doppler effect or frequency of rotor – there are no discussion.
It seems that 2-nd and 3-rd harmonics are observed during deceleration of rotor but not during other phases – why?. This effect should be discussed.
In the sentence “A total integration time of 15 s” author mentioned “integration” but it is not explained what kind of integration was used. And picture 7 shows only 9seconds.
Fig. 8 what is the difference between 8a and 8b, 8c and 8d.
Why on 8b there are two different frequencies? Why on 8a is a line around 20 Hz?, What was carrier frequency?
Similar questions can be raised to all presented results where more care is needed to describe the conditions of measurements (carrier frequency, rotor speed, length of the blades) and obtained results.
Author Response
Response to Reviewer’s Comments
We would like to thank the editor and all the reviewers of our submitted manuscript Sensors-516766, entitled “Low-Slow-Small (LSS) Target Detection based on Micro Doppler Analysis in Forward Scattering Radar Geometry". Here we have modified the paper for resubmission according to the reviewer’s suggestions and comments.
The authors have managed to answer almost all the reviewer comments and suggestions.
REVIEWER 3
General comment
Interesting paper showing both theory and measurements. Both the simulated and the measurement results are very interesting but they are not fully explained.
1) The reviewer’s comment:
“The reviewer after recommending the paper suggested some improvement in some areas; example, in Fig 3. - The model seems to be limited to the blade tips (and it should be clearly stated). In many cases also blade flashes are visible. Electromagnetic modelling will be welcome”
Response:
Firstly, we would like to thank the reviewer for the comments and important suggestion, we really appreciate that. Yes, the reviewer is right; Fig 3 is limited to response due to blade tips. This information was stated in line 214-216 of the updated version of the manuscript. It is also made clear while explaining the simulated result in line 302 and the validation in line 404-405. The electromagnetic modelling was also conducted by simulating the forward scattering RCS of three different blades type for different aspect angle by using computer simulation technology (CST). The result was interesting and was published in the “STRIDE Defence Science & Technology Journal”
2) The reviewer’s comment:
"Fig 7. Do we observe Doppler effect or frequency of rotor – there are no discussion”.
Response:
Thank you for highlighting this issue, we accept that this paragraph has some shortage in explanation regarding the Doppler effect. Actually Doppler effect due to the rotating blade was observed not the frequency of the rotor. The discussion for “Fig 7” “Now Fig 9” in the updated version of the manuscript comes before the figure as can be seen in line 452-459. For clarity, the Figure is now referenced within the text.
3) The reviewer’s comment:
“It seems that 2-nd and 3-rd harmonics are observed during deceleration of rotor but not during other phases – why? This effect should be discussed”.
Response:
Thank you for highlighting this issue, we accept that the harmonics effect was not discussed. The 2nd and 3rd harmonics were due to the volume scattering of the surrounding, and occurred while decelerating, due to the rotors free speed and the time taken to stop. This one of the motivating factor of the anechoic chamber experiment, in an attempt to minimize the harmonics and other external effect.
4) The reviewer’s comment:
“In the sentence “A total integration time of 15 s” author mentioned “integration” but it is not explained what kind of integration was used. And picture 7 shows only 9seconds”.
Response:
Thank you for highlighting this issue, we accept that this paragraph has an unclear information that may be misleading. This is an over sight, we actually refer to “observation time”. Fig 7 which is now “Fig 9” in the updated version of the manuscript shows only 9 s because the transition time of “2 s” out of “5 s” for both start and stop were captured, while the remaining “5 s” is for “On” state. This was explained in line 457-459 (the 3-states transition of the blades, i.e. Off-On-Off. The blade was initially in ‘Off’ position, then switched ‘On’ and then ‘Off’ again for 5 s each.)
5) The reviewer’s comment:
“Fig. 8 what is the difference between 8a and 8b, 8c and 8d”.
Response:
The difference between 8a and 8b, 8c and 8d which is now “Fig 10a, 10b, 10c and 10c” in the updated version of the manuscript is that, Fig. 10a is for 1-rotor and 10b for 2-rotors signatures while the blade is “Facing-Rx” scenario while Fig. 10c is for 1-rotor and 10d for 2-rotors signatures while the blade is “Facing-Up” scenario. These was stated in the Figure caption and the explanation of the Figure 10, and the explanation in line 515-520.
6) The reviewer’s comment:
“Why on 8b there are two different frequencies? Why on 8a is a line around 20 Hz? What was carrier frequency?”
Response:
The Fig 8a and 8b are now “Fig 10a and 10b” in the updated version of the manuscript. There are two different frequencies in Fig 10b because of 2-rotors with different geometrical positions as described in Fig 2 and equation (4) and (5), line 172-184. The 20 Hz line is harmonics due to reflection and this was minimized during the anechoic chamber experiment. The frequency used is 3 GHz.
7) The reviewer’s comment:
“Similar questions can be raised to all presented results where more care is needed to describe the conditions of measurements (carrier frequency, rotor speed, length of the blades) and obtained results”.
Response:
The carrier frequency used is now stated in the updated version of the manuscript for all cases e.g. line 400, line 412, line 483, line 524, line 551, line 588, line 619 etc. For the rotor speed, an approximate rotor speed of 36 r/s and blade length is 0.12 m were used; this was now stated in line 391-392.
Regards
RSA Raja Abdullah

Reviewer 4 Report
The paper reads well and is well presented.
The good paper with practical relevance with validation setup through simulation supports the intended application.
A few corrections are required:
In Fig1.a) Wd must change to wd/2.
In eq 5 and 6, the author should mention what is θo
Fig 16 and Fig 17 is better to present both results in the same span in X-axis.
Author Response
Response to Reviewer’s Comments
We would like to thank the editor and all the reviewers of our submitted manuscript Sensors-516766, entitled “Low-Slow-Small (LSS) Target Detection based on Micro Doppler Analysis in Forward Scattering Radar Geometry". Here we have modified the paper for resubmission according to the reviewer’s suggestions and comments.
The authors have managed to answer almost all the reviewer comments and suggestions.
REVIEWER 4
General comment
The paper reads well and is well presented. The good paper with practical relevance with validation setup through simulation supports the intended application. A few corrections are required:
1) The reviewer’s comment:
“In Fig1.a) Wd must change to wd/2”
Response:
Firstly, we would like to thank the reviewer for this observation, we really appreciate that. The observation was noted and corrected in the updated version “now Fig 2” of this manuscript.
2) The reviewer’s comment:
" In eq 5 and 6, the author should mention what is θo”.
Response:
Thank you for highlighting this issue also; the equations (5) and (6), “now (6) and (7)” in the updated version of this manuscript, θo is initial angle of rotation of the blade. It is the initial angle at which the blade started rotating. This is explained in the updated version of this manuscript, line 239-242.
3) The reviewer’s comment:
“Fig 16 and Fig 17 is better to present both results in the same span in X-axis”.
Response:
We appreciate your recommendation regarding Fig 16 and Fig 17 “now Fig 18 and Fig 19” in the updated manuscript. Yes, the Figures can be in the same span of X-axis; but, for clarity we decided to treat them individually regarding their discussion especially that the simulation is spread over a period of 1 s. While for the experimental STFT representation, only the signature occupies the period of 1 s.
Regards
RSA Raja Abdullah

Reviewer 5 Report
This paper has a very good idea for detecting a low SNR UAV using forward scattering radar, say, by looking Doppler shift (frequency vs time) of UAV received by the receiver. Authors want to show that the frequency-time response from the spinning of rotor blades of underlying UAV may be used as a signature to identify an UAV and even the attitude of UAV. However, the paper does not provide enough theoretical support for their claims. In particular, the following issues need to be addressed:
1. almost all equations are not explained clearly. For example, equations 1 and 2: what are the relationship between these equations and the equations presented after them (e.g. equations 3,4..) ? How the parameters in these equations are related to figure 1? Equations 3 and 4 are the coordinators of the 4 rotors. It is not clear how they are correlated to the FSR geometry in figure 2. I don't understand what information we should take from the figure 2.
2. Authors did not explain where Fig.3 comes from? It looks that figure 3b is a simulated radar return but none of the figures presented in the experiment results are similar or related to it. In addition, equation 5 and 6 are not explained, how they are related to figure 2?
3. Many experiment results are presented in the paper. But I would argue that the experiment scenario is not adequate, that is, the test distance between radar and UAV is too short. Within a short distance, what is the advantage of using the FSR over a video camera for UAV detection?
While the paper has enough context in length, it has not written technically sound. Poor technical justification and derivation are through out the paper.
I therefore suggest to do a major revision by addressing the above issues.
Author Response
Response to Reviewer’s Comments
We would like to thank the editor and all the reviewers of our submitted manuscript Sensors-516766, entitled “Low-Slow-Small (LSS) Target Detection based on Micro Doppler Analysis in Forward Scattering Radar Geometry". Here we have modified the paper for resubmission according to the reviewer’s suggestions and comments.
The authors have managed to answer almost all the reviewer comments and suggestions.
REVIEWER 5
General comment
This paper has a very good idea for detecting a low SNR UAV using forward scattering radar, say, by looking Doppler shift (frequency vs time) of UAV received by the receiver. Authors want to show that the frequency-time response from the spinning of rotor blades of underlying UAV may be used as a signature to identify an UAV and even the attitude of UAV. However, the paper does not provide enough theoretical support for their claims. In particular, the following issues need to be addressed:
1) The reviewer’s comment:
“The reviewer suggests to better clarify the equations and how they are used in generating the Doppler signature. Almost all equations are not explained clearly. For example, equations 1 and 2: what are the relationship between these equations and the equations presented after them (e.g. equations 3, 4..) ? How the parameters in these equations are related to figure 1? Equations 3 and 4 are the coordinators of the 4 rotors. It is not clear how they are correlated to the FSR geometry in Figure 2. I don't understand what information we should take from the Figure 2”.
Response:
Firstly, we would like to thank the reviewer for the comments and important suggestion, we really appreciate that. Yes, the reviewer is right; the equations are not properly explained. An is therefore made in the updated version for more simplified and clear understanding of the usage of the equations and figures.
For the derivation, to make it clearer, we have added new figure “Figure 1” to describe how the signal was initiated. Equation (1) Srx = Adir Cos(ωc)t + Asc Sin(ωc + ωsc)t represented the entire received signal which is a combination of direct and scattered signal. The manuscript explains further on the micro–Doppler within the received signal as describe in equation (2) – (13).
This equation was used to generate a Doppler effect due to the drone radial motion; as can be seen in Fig 1 of the updated version of the manuscript.
Within the scattered signal ωsc = 2πfmd, a Doppler component ”fmd“ is present; this Doppler component is used in equation (12) such that . For the velocity due to the blade-tip “Vtip”; this was defined in equation 6, line 218.
Equation (3) and (4) which are now “Equations (4) and (5)” are the vectors representing each rotor geometrical position in Figure 2, while Fig 3 is a representation of 1-rotor with 2-blades. Each rotor was treated independently referenced to its rotor vectors as defined by equations (3) and (4) i.e. “rotor geometrical position” of Fig 2. Figure 3 was used to generate the transmitter and receiver ranges; these ranges are used to calculate the bistatic angle of the blade-tip “βtip” while the blade is rotating at a velocity “Vtip”. The updated version of this manuscript contained a more detailed and clear explanation from line 98-264.
2) The reviewer’s comment:
“Authors did not explain where Fig.3 comes from? It looks that figure 3b is a simulated radar return but none of the figures presented in the experiment results are similar or related to it. In addition, equation 5 and 6 are not explained, how they are related to figure 2?”
Response:
Fig 3 which is now “Figure 4” was generated from Fig 2 and Fig 3 as stated in line 266-269. It is the simulated results based on the theoretical assumptions of Fig 2 and 3, and also equations 1 - 15. All these were based on the actual drone geometrical dimensions and operational parameters, in addition to that of the FSR geometry. a detailed explanation is now added in the updated version of this manuscript (See line 266-276). The signature comparison between the simulated and the experimental was now presented in the updated version of this manuscript as presented by Fig 7 and explained in line 419-430.
For equations (5) and (6) “now (6) and (7)” are the velocity due to blade-tip “Vtip” used in equation 12, i.e. and the range equation made by transmitter-to-blade-tip and blade-tip-to-receiver, defined by Rrx_tip_nm of rotor “n”, blade “m”. Detail explanation of the relationship between equation (6) and (7) is now explained in line 208-230.
3) The reviewer’s comment:
“Many experiment results are presented in the paper. But I would argue that the experiment scenario is not adequate, that is, the test distance between radar and UAV is too short. Within a short distance, what is the advantage of using the FSR over a video camera for UAV detection?”
Response:
Thank you for highlighting this issue, we accept that some experiment result had some issues, yet, effort was made appropriately maintained all scenarios aforementioned in the theoretical design. However, it was stated in line 414–416 that the Fig 8 was not to scale but rather for illustration purposes. For section 4.3, due to the limitation of the max achievable transmitter height, a very low transmitting power was used to observe the FSR enhanced capability of detecting the micro-motions. Therefore, the system capability is not limited to the achieved distance.
We try as much as possible to imitate the Satellite by using low power transmit and also the actual satellite dish receiver with narrow beamwdith. We considered FSR system over camera to minimize the challenges suffered by camera based method such as weather effect, dynamic or blurred background among others. The radar system was considered as an alternative to dark, noisy and blurred or misty environments. This was also contained in the main text line 55-56 and line 672-674.
4) The reviewer’s comment:
“While the paper has enough context in length, it has not written technically sound. Poor technical justification and derivation are throughout the paper. I therefore suggest to do a major revision by addressing the above issues”.
Response:
Thank you for highlighting this issue, and we accept that the paper previously lacks some technical justifications. We therefore put more effort and address this issue in the updated version of this manuscript. Effort involving more figures and derivations was made to improve the technical content of this work. Equations were correlated with the figures thus, given the reader a clear view of the entire content. These modifications can be seen from line 98-313, line 319-324 etc. All the assumptions were made based on the real commercial drone parameters and geometry.
Regards
RSA Raja Abdullah

Round 2
Reviewer 5 Report
The revised version has significant improvement compared to the initial submission. I can see that both technical quality and method justification are enhanced. I am satisfied with the authors response to my comments and thus I accept it for publication.